# Accidental hypothermia in emergency care: multifactorial triage-based prediction of early critical outcomes in a temperate-climate cohort

Kornél Ádám[1]*, Anna Stelkovics[1], Barbara Zadravecz-Heider[1], Dóra Melicher[1], Zsolt Bognár[2], Barbara V. Farkas[1], Bánk G. Fenyves[1], Szabolcs Gaál-Marschal[1,3], Csaba Varga[1,4]

1 Department of Emergency Medicine, Semmelweis University, Budapest, Hungary, 2 Department of Emergency Medicine, Heim Pál National Pediatric Institute, Budapest, Hungary, 3 Department of Emergency Medicine, North-Buda St. John Central Hospital, Budapest, Hungary, 4 Department of Emergency Medicine, Kaposi Mór Teaching Hospital, Kaposvár, Hungary

* adam.kornel@semmelweis.hu

**Citation:** Ádám K, Stelkovics A, Zadravecz-Heider B, Melicher D, Bognár Z, Farkas BV, et al. (2025) Accidental hypothermia in emergency care: multifactorial triage-based prediction of early critical outcomes in a temperate-climate cohort. PLoS One 20(10): e0334328. https://doi.org/10.1371/journal.pone.0334328

## Abstract

### Background

Accidental hypothermia, defined by a core temperature <35 °C destabilizes metabolism, ventilation, and circulation, precipitating malignant arrhythmias or cardiac arrest. We characterized such patients in a Hungarian emergency department and sought early predictors of death or intensive care requirement.

### Methods

In this retrospective cohort (2020–2024) at Semmelweis University, adults with measured tympanic temperature <35 °C were identified. Demographics and Hungarian Emergency Triage System categories were recorded. Severity was graded based on the thresholds of Swiss staging and the Wilderness Medical Society classification. The primary outcome was emergency department death or admission to the intensive care unit. Prognostic performance of triage category, admission temperature, hypothermia severity thresholds, and combined models was assessed with receiver-operating-characteristic analysis. Odds ratios were derived from logistic regression, and separate receiver-operating-characteristic curves were generated for each predictor.

### Results

A total of 131 patients met the inclusion criteria. Median age was 67.0 years (IQR 59.0–75.0); 88 (67.2%) were male. Median admission temperature was 29.3 °C (IQR 26.1–31.4 °C); 47 (35.9%) had severe hypothermia (<28 °C). Median emergency department length of stay was 13.7 h (IQR 9.5–18.9 h). Sixteen patients (12.2%) required intensive care and 28 (21.4%) died before transfer, yielding a 33.6%

**Data availability statement:** All relevant de-identified data that support the findings of this study are publicly available via the Open Science Framework at https://doi.org/10.17605/OSF.IO/5URSV.

**Funding:** The author(s) received no specific funding for this work.

**Competing interests:** The authors have declared that no competing interests exist.

**Abbreviations:** AUC, area under the ROC curve; AVPU, alert, verbal, pain, unresponsive (neurological responsiveness scale); ATS, Australasian Triage Scale; CI, confidence interval; Cramer's V, Measure of association for nominal variables; CTAS, Canadian Triage and Acuity Scale; ED, emergency department; ICD-10, International Classification of Diseases, Tenth Revision; ICU, intensive care unit; IQR, interquartile range; LR+, Positive likelihood ratio; LR-, Negative likelihood ratio; MSTR, Hungarian Emergency Triage System (Magyar Sürgősségi Triage Rendszer); MTS, Manchester Triage System; n, number (of patients or observations); OR, odds ratio; p value (p), probability value (statistical significance indicator); R² (Cox-Snell, Nagelkerke), coefficient(s) of determination in logistic regression; ROC, receiver operating characteristic (curve); SD, standard deviation; SPSS, Statistical Package for the Social Sciences; T68, ICD-10 code for hypothermia; Tymp., Tympanic (temperature measurement); VIF, Variance inflation factor; WMS, Wilderness Medical Society

critical-outcome rate (44/131). Incidence tracked seasonal ambient temperatures, yet environmental temperature itself was not associated with the composite outcome. Triage category predicted critical outcome better than thresholds of either hypothermia-specific scale (AUC 0.683). Adding admission temperature improved accuracy (AUC 0.740, 95% CI 0.644–0.829).

## Conclusions

Despite milder winters, accidental hypothermia still carries substantial early mortality. Integrating admission temperature into a general triage system enhances prognostication and may guide rapid escalation of care. Our findings suggest the need for systematic surveillance, focused clinician education, and targeted resources to protect vulnerable patients in increasingly variable climates.

---

## Background

Accidental hypothermia is defined as a drop in core temperature below 35°C. If untreated, it impairs metabolic processes, induces respiratory and circulatory depression, and may lead to life-threatening arrhythmias and cardiac arrest. It may develop despite intact thermoregulation due to cold exposure or pathological thermoregulatory failure [1]. In winter and early spring, the incidence increases [2,3]. As global climate change reduces extended cold periods, strong, brief cold spells may increase, placing older, chronically ill, and socially marginalized people in danger [4,5].

Two tools are generally used for hypothermia staging, when direct core temperature is not available (Table 1). The Swiss staging model combines physiological responses and symptomatology with predefined core temperature ranges [6]. A revised version estimates circulatory-arrest risk when core temperature is not available [7]. The field-use Wilderness Medical Society classification even offers guidance on rewarming techniques for each severity level [8].

Emergency triage systems assess hypothermia in different ways, prioritizing patients by evaluating the urgency of their condition, frequently employing five-category scales [9]. Despite this common foundation, these systems differ in the detailed criteria they use for patient assessment and coexisting specific medical conditions, also in terms of hypothermia (Table 2).

In the Manchester Triage System (MTS), a core temperature below 35°C is a modifier for category 2, but the final decision is made with additional symptoms and clinical factors considered. [10]. Hungary used a modified version of the widely used, multidimensional Canadian Triage and Acuity Scale (CTAS) [11]. In the CTAS-based Hungarian Emergency Triage System (MSTR), a core temperature between 32°C and 35°C is sufficient for classification into category 3, whereas a core temperature less than 32°C is an independent modifier for assigning patients to category 2 (S1–S2 Tables), indicating the need for immediate intervention [11, 12]. Croatia is among the countries that have implemented the Australasian Triage Scale, despite its rarity in Europe. It places life-threatening conditions above the severity of hypothermia on

**Table 1. Hypothermia severity classification systems [6–8].**

| Swiss staging model for hypothermia | | | | | Wilderness Medical Society classification |
|---|---|---|---|---|---|
| **Original** | | **Revised** | | | |
| Clinical findings | Estimated core temperature (°C) | AVPU | Risk of hypothermic cardiac arrest | Category and clinical findings | Estimated core temperature (°C) |
| Clear consciousness with shivering | 35–32 | "Alert" | Low | Mild: Normal mental status, shivering, but not functioning normally and unable to care for self | 35–32 |
| Impaired consciousness without shivering | <32–28 | "Verbal" from AVPU | Moderate | Moderate: Abnormal mental status with shivering, or abnormal mental status without shivering, but conscious | 32–28 |
| Unconsciousness | <28–24 | "Painful" or "Unconscious" from AVPU and vital signs | High | Severe/profound: Unconscious | < 28°C |
| Apparent death | <24–13.7 | "Unconscious'" from AVPU and no detectable vital signs | Hypothermic cardiac arrest | | |
| Death due to irreversible hypothermia | <13.7? (<9?) | | | | |

Note: The "Stage V" is a historical/literature concept and is not part of the current clinical staging.

**Table 2. Five-level emergency triage systems [9–13].**

| Levels | System | Category | Color | Time to treat | Hypothermia modifying threshold | Clinical characteristics |
|---|---|---|---|---|---|---|
| **Level I** | ATS | Resuscitation | Red | Immediate | – | Immediate life-threat (airway, breathing, or circulation compromise); cardiac arrest if present |
| | CTAS | | | | | |
| | MTS | Immediate | | | | |
| **Level II** | ATS | Emergency | Orange | ≤10 min. | – | Impending life threat or condition requiring time critical intervention. |
| | CTAS | Emergent | | ≤15 min. | <32°C | |
| | MTS | Very Urgent | | ≤10 min. | <35°C | |
| **Level III** | ATS | Urgent | Yellow | ≤30 min. | – | Potential life threat or situational emergency requiring symptom management. |
| | CTAS | | | | 35 − 32°C | |
| | MTS | | | ≤60 min. | – | |
| **Level IV** | ATS | Semi-Urgent | Blue | ≤60 min. | – | Potentially severe conditions with complexity and urgency. |
| | CTAS | Less Urgent | Green | | | |
| | | MTS | Standard | ≤120 min. | | |
| **Level V** | ATS | Non-Urgent | White | ≤120 min. | – | Less urgent conditions, generally not life-threatening. |
| | CTAS | | | | | |
| | MTS | | Blue | ≤240 min. | | |

ATS: Australasian Triage Scale; CTAS: Canadian Triage and Acuity Scale; MTS: Manchester Triage System; "–" indicates no hypothermia modifier.

its own. [13–15]. When core temperature is a unique modifier, triage system criteria may affect categorization and even treatment [9]. Demographics and critical outcomes may serve as a foundation for local procedure adaptation [16].

In real-world emergency care, the predictive accuracy and clinical relevance of specialized hypothermia classification systems—along with conventional triage protocols applied to hypothermic patients—remain insufficiently documented. Furthermore, epidemiological data on hypothermia in Hungary, including seasonal variation, patient demographics, and

emergency department outcomes, are notably scarce. To address these gaps, we conducted a retrospective study of hypothermic patients treated at the Department of Emergency Medicine, Semmelweis University, between 2020 and 2024. Our objectives were as follows:

- Analyze the demographic characteristics and seasonal trends of hypothermia cases.

- Evaluate the predictive performance of the Hungarian Emergency Triage System – as it is the nationally mandated, CTAS-based protocol in use across Hungary – and of the severity categories defined by measured tympanic temperature thresholds in the Swiss staging model and Wilderness Medical Society guidelines for identifying patients at risk of critical outcomes.

- Assessing whether incorporating the admission tympanic temperature into the MSTR improves risk stratification.

-  Investigate the impact of outdoor temperature on hypothermic patient outcomes

## Methods

A retrospective cohort analysis was conducted to identify patients presenting with hypothermia to the Department of Emergency Medicine, Semmelweis University, between January 1, 2020, and December 31, 2024. During this period, a total of 197 751 emergency department visits occurred (31 264 in 2020; 41 571 in 2021; 38 777 in 2022; 39 771 in 2023; and 46 368 in 2024). Hungarian emergency departments are required to operate at least one ICU-equipped resuscitation bay and maintain 4–8 monitored beds, including at least one capable of intensive-level care [17]. The inclusion criteria were ICD-10 code T68 and a tympanic temperature below 35.0°C at ED admission, which was manually retrieved from the EHR on January 5, 2025 (eMedSolution system database version 2024/Q4/1). Patients with a tympanic temperature of 35.0°C or higher and those with incomplete data were excluded. (Fig 1). Ethical approval for the study was granted by the Semmelweis University Regional and Institutional Review Board (SE RKEB 274/2024). Only fully de-identified, retrospectively collected clinical records were accessed and analyzed. Consequently, individual informed consent was waived.

Demographic data (age, sex, residence) and clinical characteristics (mode of arrival, MSTR triage category, hypothermia severity) were recorded. Categorical variables are reported as frequencies and percentages, age as the median (IQR), and tympanic temperature as the median (IQR). Hypothermia severity was categorized via the Swiss staging model and the Wilderness Medical Society classification based on standardized measurements of tympanic temperature ranges at admission. Admission tympanic temperature used as a pragmatic proxy for core temperature was assessed using Braun ThermoScan PRO 6000 tympanic thermometer, which uses infrared technology (Hillrom Inc.) [18]. The instrument displays values between 20 °C and 42.2 °C, clinical accuracy is ± 0.2 °C within the 35.0-42.0 °C range and ±0.3 °C outside this interval. These limits conform to ASTM E 1965−98 requirements for infrared thermometers (S3 Table). Hypothermia severity was then categorized according to the ranges specified in the Swiss Staging System and the Wilderness Medical Society classification. Ambient temperature was exported from the monthly mean temperature data from the local meteorological station based on the registered residence of the patients (S4 Table) [19]. Data normality was evaluated via the Kolmogorov–Smirnov and Shapiro–Wilk tests.

Critical outcomes were defined as mortality during emergency care or admission to the ICU to capture all critical events requiring intensive care during the ED stay and to reflect the clinical relevance of both early mortality and the need for high-level intervention in a setting with limited intensive care availability. Relationships among categorical variables were assessed via the chi-square test. The effect size was quantified via Cramer's V. The Mann–Whitney U test and rank–biserial correlation coefficients were implemented for continuous data that were not normally distributed. Pearson's correlation coefficient was used to assess the association between normally distributed data. Univariate logistic regression models were implemented to evaluate the correlations of the triage category, admission tympanic temperature, WMS, Swiss staging, and ambient temperature with the primary outcome. A multivariate logistic regression model was employed to

 

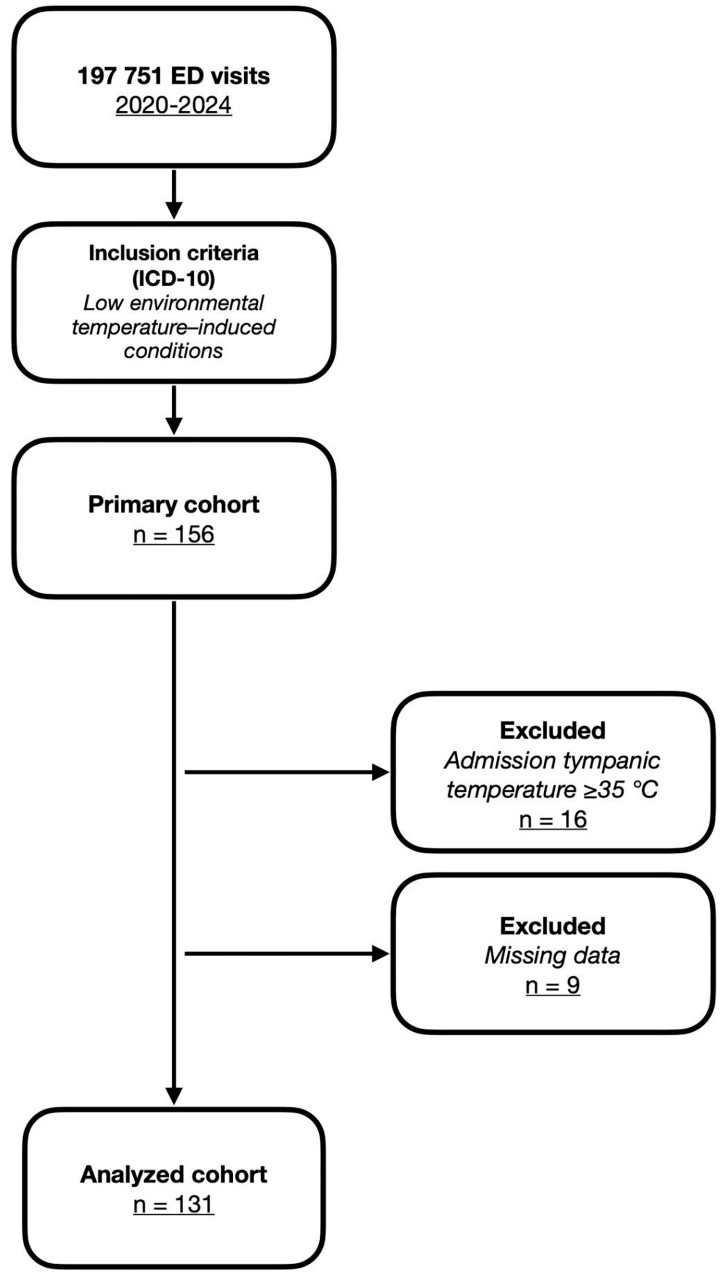

**Fig 1. Patient selection flowchart.**

calculate odds ratios with 95% confidence intervals, p values, and Wald statistics for components with low collinearity, as determined by the variance inflation factor (VIF). We further assessed potential collinearity between triage category (MSTR) and admission tympanic temperature using Pearson's and Spearman's correlations in combined models (S7 Table). In the multivariate models, we evaluated the predictive accuracy and clinical relevance of combinations of predictors that were independently significant for the critical outcomes. Variables without significant univariate association were not included in the multivariate models.

We used multiple model validity measures to conduct a thorough assessment of the predictive performance and model fit. A high-risk indicator for critical outcomes was determined by combining the triage category and tympanic admission temperature in a logistic regression model. Calibration was evaluated using the Brier score, intersection point, slope, and visual inspection with decile grouping and a locally weighted scatterplot smoothing (LOWESS) curve. The probability threshold was set based on the Youden index. We report odds ratios with 95% confidence intervals and Nagelkerke $R^2$ from logistic models, and PPV, NPV, sensitivity, specificity, LR+, LR−, and AUC with 95% CIs at the Youden-optimal threshold. Analyses were performed in SPSS 28.0 and R 4.2.0 with $p < 0.05$.

All relevant de-identified data that support the findings of this study are publicly available via the Open Science Framework at https://doi.org/10.17605/OSF.IO/5URSV. The public dataset removes all direct and indirect identifiers in accordance with PLOS ONE's data policy on human participant data.

## Results

### Study population and demographics

Of the 131 patients, n = 88 (67.2%) were male. The median age was 67.0 years (IQR: 59.0–75.0). A total of 74 (56.5%) patients were aged ≥65 years (18–44: 4 [3.1%]; 45–64: 53 [40.5%]) (Table 3).

### Age and triage classification

Among the hypothermic patients, 129 (98.5%) arrived by ambulance. n = 61 (46.6%) were classified as the most urgent triage category I (resuscitation), n = 58 (44.3%) as category II (emergent), and n = 12 (9.1%) as less urgent categories III-IV (urgent and less urgent), respectively, on the basis of the severity of their condition upon arrival. Despite the high proportion of category I presentations, no patient in the cohort arrived in cardiopulmonary arrest. No patient was classified as MSTR category V. Via tympanic thermometry, the median temperature upon arrival was 29.3°C (IQR: 26.1–31.4°C), suggesting severe hypothermia in a substantial number of patients (Table 3).

### Distribution of hypothermia by severity

Based on the Swiss Staging System, 29 (22.1%) patients had stage I hypothermia, 55 (42.0%) stage II, 37 (28.2%) stage III, and 10 (7.6%) stage IV. According to the Wilderness Medical Society classification, n = 29 (22.1%) participants presented with mild hypothermia, n = 55 (42.0%) with moderate hypothermia, and n = 47 (35.9%) with severe hypothermia, which is largely consistent with the distribution according to the Swiss counterpart (Table 3).

### Seasonal variation

When the monthly mean temperature dropped below 7°C in winter, the incidence of hypothermia increased across all severity stages. All Swiss staging categories (I–IV) occurred at temperatures below 10°C, with more than half of Stage II–IV cases occurring in months with a mean temperature less than 7°C. Stage IV hypothermia (<24°C) was the most temperature dependent, with 90.0% (9/10) of the cases occurring below 10°C and none occurring above 18°C (Fig 2).

Total ED visits rose from 31 264 in 2020–46 368 in 2024. Mean monthly ambient temperatures varied between 0.9 °C and 26.6 °C over the same period (S4 Table).

### Length of stay and outcomes

The median length of ED stay was 13.7 h (IQR 9.5–18.9). A total of 16 (12.2%) patients were admitted to the intensive care unit, and 81 (61.8%) were transferred to inpatient wards. The discharge rate was n = 6 (4.6%), and the mortality rate was n = 28 (21.4%) (Table 3).

**Table 3. Demographics, clinical characteristics, and outcomes of emergency care.**

| | 2020 | 2021 | 2022 | 2023 | 2024 | 2020–24 |
|---|---|---|---|---|---|---|
| **Gender** | | | | | | |
| Male % (n) | 59.1 (13) | 70.0 (14) | 63.6 (14) | 69.2 (18) | 70.7 (29) | 67.2 (88) |
| Female % (n) | 40.9 (9) | 30.0 (6) | 36.4 (8) | 30.8 (8) | 29.3 (12) | 32.8 (43) |
| Total (n) | 22 | 20 | 22 | 26 | 41 | 131 |
| **Age** | | | | | | |
| Median (IQR) | 69.0 (60.5-76.8) | 70.0 (62.5-75.0) | 65.0 (57.0-68.0) | 72.0 (63.5-79.0) | 64.0 (57.0-74.0) | 67.0 (59.0-75.0) |
| 18–44 years % (n) | 0.0 (0) | 0.0 (0) | 4.5 (1) | 3.8 (1) | 4.8 (2) | 3.1 (4) |
| 45–64 years % (n) | 40.9 (9) | 35.0 (7) | 40.9 (9) | 23.1 (6) | 53.7 (22) | 40.5 (53) |
| ≥65 years % (n) | 59.1 (13) | 65.0 (13) | 54.6 (12) | 73.1 (19) | 41.5 (17) | 56.5 (74) |
| **Form of arrival** | | | | | | |
| By ambulance % (n) | 100.0 (22) | 100.0 (20 | 100.0 (22) | 92.3 (24) | 100.0 (41) | 98.5 (129) |
| **Triage category** | | | | | | |
| MSTR I % (n) | 45.5 (10) | 65.0 (13) | 54.5 (12) | 23.1 (6) | 48.8 (20) | 46.6 (61) |
| MSTR II % (n) | 54.5 (12) | 35.0 (7) | 45.5 (10) | 57.7 (15) | 34.1 (14) | 44.3 (58) |
| MSTR III % (n) | 0.0 (0) | 0.0 (0) | 0.0 (0) | 15.4 (4) | 9.8 (4) | 6.1 (8) |
| MSTR IV % (n) | 0.0 (0) | 0.0 (0) | 0.0 (0) | 3.8 (1) | 7.3 (3) | 3.0 (4) |
| MSTR V % (n) | – | – | – | – | – | – |
| **Tympanic (core-proxy) temperature at arrival** | | | | | | |
| Tymp. °C median (IQR) | 29.5 (25.4-32.3) | 29.1 (25.9-31.3) | 29.4 (27.8-31.5) | 30.1 (27.6-32.1) | 29.1 (27.0-30.0) | 29.3 (26.1-31.4) |
| **Swiss staging model for hypothermia** | | | | | | |
| Stage I % (n) | 31.8 (7) | 20.0 (4) | 22.7 (5) | 26.9 (7) | 14.6 (6) | 22.1 (29) |
| Stage II % (n) | 22.7 (5) | 45.0 (9) | 50.0 (11) | 42.3 (11) | 46.3 (19) | 42.0 (55) |
| Stage III % (n) | 40.9 (9) | 25.0 (5) | 22.7 (5) | 23.1 (6) | 29.3 (12) | 28.2 (37) |
| Stage IV % (n) | 4.5 (1) | 10.0 (2) | 4.5 (1) | 7.7 (2) | 9.8 (4) | 7.6 (10) |
| Stage V % (n) | – | – | – | – | – | – |
| **Wilderness Medical Society classification** | | | | | | |
| Mild % (n) | 31.8 (7) | 20.0 (4) | 22.7 (5) | 26.9 (7) | 14.6 (6) | 22.1 (29) |
| Moderate % (n) | 22.7 (5) | 45.0 (9) | 50.0 (11) | 42.3 (11) | 46.3 (19) | 42.0 (55) |
| Severe % (n) | 45.5 (10) | 35.0 (7) | 27.3 (6) | 30.8 (8) | 39.1 (16) | 35.9 (47) |

*(Continued)*

**Table 3.** (Continued)

|  | 2020 | 2021 | 2022 | 2023 | 2024 | 2020–24 |
|---|---|---|---|---|---|---|
| **Care and outcome** | | | | | | |
| Median length of stay (IQR) | 10.2 (7.3-23.6) | 16.3 (12.4-20.1) | 16.1 (11.6-17.9) | 15.8 (11.1-20.6) | 11.8 (8.9-15.0) | 13.7 (9.5-18.9) |
| Discharged % (n) | 0.0 (0) | 5.0 (1) | 4.5 (1) | 15.4 (4) | 0.0 (0) | 4.6 (6) |
| Wards % (n) | 59.1 (13) | 70.0 (14) | 54.6 (12) | 50.0 (13) | 70.8 (29) | 61.8 (81) |
| ICU % (n) | 13.6 (3) | 5.0 (1) | 13.6 (3) | 11.5 (3) | 14.6 (6) | 12.2 (16) |
| Died in ED % (n) | 27.3 (6) | 20.0 (4) | 27.3 (6) | 23.1 (6) | 14.6 (6) | 21.4 (28) |

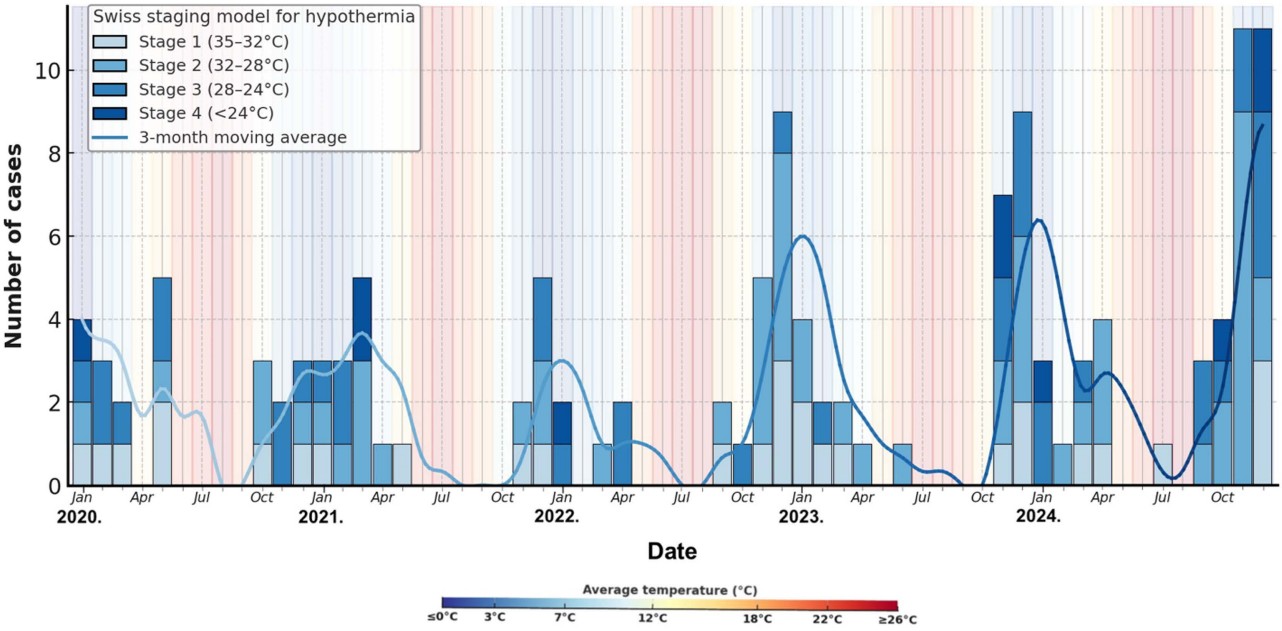

**Fig 2. Seasonal distribution of hypothermia cases and ambient temperatures (2020–2024). Bars indicate monthly case counts stratified by Swiss staging (I: 35–32 °C; II: 32–28 °C; III: 28–24 °C; IV: <24 °C); line shows the monthly mean outdoor temperature for patients' registered area. Sample size: n = 131. Meteorological source: monthly means from the local station (S4 Table).**

## Model performance and collinearity

Triage and admission temperature were only modestly correlated (Spearman ρ = 0.309; Pearson r = 0.265). In the combined triage+temperature model, VIFs were 1.075 for both variables. (S7 Table)

## Predictors and univariate correlations with severe outcomes

The triage category was the strongest predictor (p < 0.001), indicating a medium–strong association (Cramer's V = 0.349) with the primary outcome. The temperature ranges from the Swiss Staging System also showed a significant, but moderate-strength association with the primary outcome (p = 0.032; Cramer's V = 0.259). There was also a moderate but

significant association for the WMS classification (p = 0.017; Cramer's V = 0.249). The initial tympanic temperature was negatively correlated (p < 0.001; Pearson's r = −0.294) with ICU admission and ED death. In contrast, ambient external temperature was not significantly associated with our outcome measure (p = 0.694; r = −0.034) (Table 4).

## Multivariate logistic regression analysis

Each one-level increase in MSTR triage category (I → II → III → IV) was associated with a ~69% reduction in the odds of a critical outcome (OR = 0.310; 95% CI 0.157–0.610; p < 0.001). Each 1 °C increase in admission tympanic temperature reduced the odds by 17.2% (OR = 0.828; 95% CI 0.738–0.928; p = 0.001). The risk significantly increased with increasing stage, as indicated by Swiss hypothermia staging thresholds (OR = 1.877, 95% CI: 1.208–2.919, p = 0.005). Moreover, the WMS classification thresholds (OR = 2.035, 95% CI: 1.203–3.442, p = 0.008) indicated a more than twofold increase in the likelihood of ICU admission or ED mortality (S5 Table).

## Combined models

Improved predictive accuracy was observed in the bivariate models. Compared with any of the univariate models, the combination of tympanic temperature and triage category (OR = 0.390, 95% CI: 0.197–0.774, p = 0.007; Nagelkerke R² = 0.199) offered significantly more precise predictions. Similarly, the results of the combination of Swiss classification and triage category (OR = 0.372, 95% CI: 0.189–0.733, p = 0.004; Nagelkerke R² = 0.183) were comparable (S5 Table).

## Predictive performance

The diagnostic accuracy of predicting critical outcomes was assessed via receiver operating characteristic (ROC) curves. Among the triage category (AUC = 0.683, 95% CI: 0.598–0.767), initial tympanic temperature (AUC = 0.666, 95% CI: 0.566–0.764), Swiss- (AUC = 0.644, 95% CI: 0.545–0.736) and WMS classification (AUC = 0.637, 95% CI: 0.536–0.730), the triage category was found to be the strongest independent predictor (Fig 3A), but the combination of triage classification and tympanic temperature (AUC = 0.740, 95% CI: 0.644–0.829) provided the best predictive performance for critical outcomes (Fig 3B). At the Youden-optimal threshold, the combined triage + temperature model yielded sensitivity 0.795 and specificity 0.621, corresponding to LR+ = 2.10 and LR− = 0.33. The model showed good overall accuracy (Brier 0.188 vs. null 0.223) and near-ideal calibration (intercept ~ 0, slope ~ 1). At the Youden-optimal threshold (p ≥ 0.282), this is a simple, high-risk signal (S1 Fig; S6 Table).

## Discussion

Consistent with earlier epidemiological findings, our data confirm that hypothermia cases in emergency care follow a seasonal trend, reaching their peak in winter [4]. Hypothermia cases were at their highest in the winter months. While the number of patients treated in the ED also increased throughout the years of the study, our dataset does not allow us to analyze if the increase in patient throughput had an independent effect on the incidence of hypothermia. Hence, we

**Table 4. Association between predictors and critical outcomes (ICU admission and ED mortality).**

| Predictor | p value | Effect size (Cramer's V/r) |
| --- | --- | --- |
| Triage category | < 0.001 | 0.349 |
| Swiss staging model for hypothermia | 0.032 | 0.259 |
| Wilderness Medical Society classification | 0.017 | 0.249 |
| Tympanic temperature at arrival (°C) | < 0.001 | −0.294 |
| Average outdoor temperature (°C) | 0.694 | −0.034 |

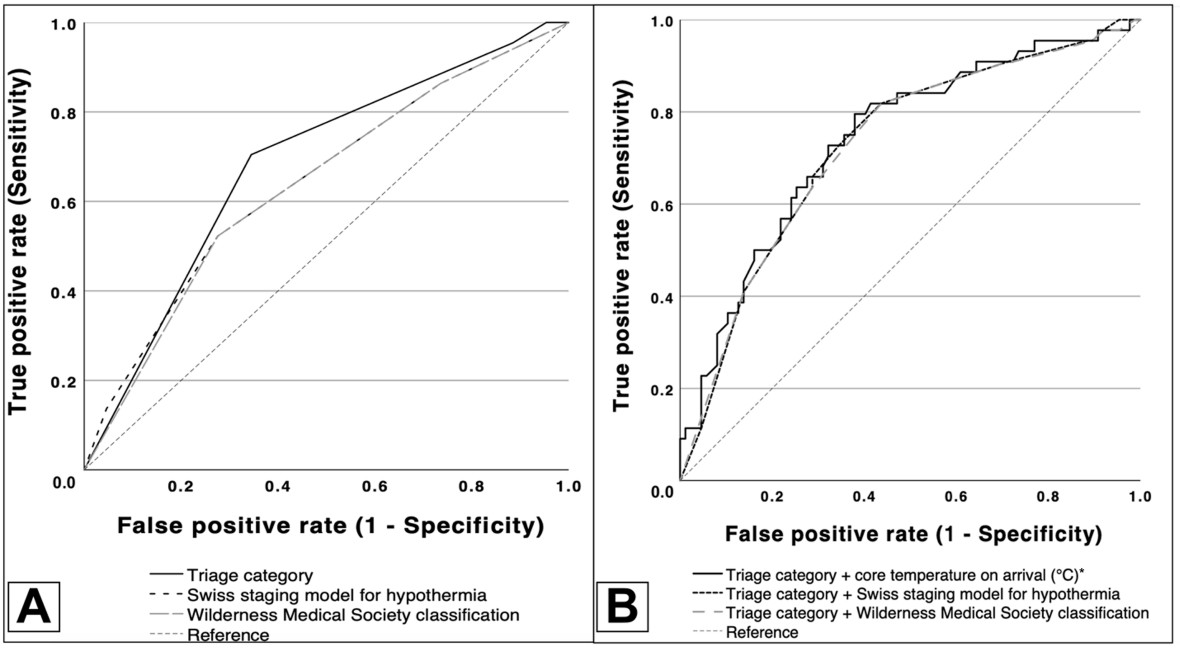

**Fig 3. ROC curves for predictors of the composite critical outcome (ICU admission or ED mortality).** Curves are shown for MSTR triage category, admission tympanic (*core-proxy) temperature (°C), Swiss staging, Wilderness Medical Society (WMS) classification, and the combined triage+temperature model. AUC values showing overall discrimination (higher is better). Abbreviations: AUC, area under the ROC curve; MSTR, Hungarian Emergency Triage System; WMS, Wilderness Medical Society.

choose not to make any causal statements. The greater proportion of male patients is also consistent with prior research, which links this trend to differences in exposure and behavioral factors [16]. The highest occurrence was seen during the cold and winter period. The severity of the cases could not be predicted by the average monthly temperature, probably indicating that patients were exposed to different environmental conditions: frailty, comorbidity, intoxication, indoor occurrence, and pre-hospital times. All of these factors influence outcomes far beyond the ambient temperature. Preventive measures should be continual, ensuring adequate readiness even before the cold season begins.

The age distribution of the patients indicated that middle-aged individuals were also prominently represented, whereas hypothermia primarily affected the older age group, as previously reported in large-scale studies [20]. Although comorbidities and age-related decreases in thermoregulatory ability are significant variables, hypothermia may also be caused by non-physiological factors [5]. Recent findings show distinct comorbidity and cognitive decline patterns between indoor and outdoor hypothermia mortality. [21].

The significance of prehospital detection and timely intervention is well documented, particularly in the context of hypothermia, where rapid recognition and management can enhance outcomes [22]. The early recognition of hemodynamic instability and the use of advanced rewarming methods improves outcomes [23]. Even the patients in unwitnessed arrest, asystole, or at old age may benefit from extracorporeal support [24]. Most of our patients arrived via ambulances, representing the critical importance of emergency medical services in hypothermia. There were no hypothermic arrests on arrival, suggesting that local EMS already transports these patients to ECMO centers, in line with European guidelines [25]. However, prehospital destination policies were not captured in the study. Emergency triage systems are tools to prioritize urgency, and they are not meant to predict outcomes. The CTAS-based Hungarian Triage System (MSTR) is, however, multimodal. It synthesizes vital-sign thresholds, mental status and signs of shock, and treats core temperature as a secondary modifier (<32 °C→MSTR II; 32–35 °C→MSTR III). Admission temperature might capture exposure and when

compared to triage, it is only modestly correlated, even in hypothermic patients. Adding the two together, may serve as complementary information. The predictive function of decision-making systems in emergency treatment was substantiated by the strong association between critical outcomes and triage categorization. The MSTR exhibited greater accuracy for the primary outcome than hypothermia-specific categorization methods did. Beyond discriminatory power, calibration analysis also showed that the triage+temperature model appears to be reliable for estimating the risk of critical outcomes. This correlation could serve as the basis for developing a simple signaling system to aid early escalation decisions in the ED.

Although their predictive power was not as strong as that of the triage classification, the temperature ranges of Wilderness Medical Society classification and the Swiss staging model for hypothermia both correlated with critical outcomes. Earlier symptom-based reports found only about 50% match between clinical stage and centrally measured core temperature [26], while a later study with higher case count showed 61% correct classification by Swiss temperature thresholds (18% overestimation; 21% underestimation) [27]. Based on this, the Swiss staging model and other hypothermia classification systems are good for determining severity, the involvement of additional clinical parameters within triage systems may aid in risk stratification and guide treatment in emergency environments [8].

Critical outcomes were associated with lower tympanic temperatures. Myocardial irritability below 28°C is a known key driver of rescue collapse and mortality in severe hypothermia [28]. Only 12.2% of patients were admitted to the ICU, despite a 21% in-ED mortality rate. Limited ICU capacity may shift critical care to the ED and make the combined outcome a better reflection of real-world severity. In isolation however, the tympanic temperature showed only moderate standalone discrimination. The absence of a significant correlation between the primary outcome and outside temperature aligns with the hypothesis that the severity of hypothermia is influenced by exposure duration, comorbidities, and other environmental variables, in addition to ambient temperature. Combining triage with the admission tympanic temperature results in a more accurate prediction than does depending on only a few factors. Our results imply that advanced risk assessment models in emergency care may perform better than single clinical indicators do and may improve patient management and resource allocation [29]. Frailty, acidosis, hyperkalemia, and indoor occurrence are known predictors of poor outcomes in hypothermia [30], which may be integrated into future severity scales.

## Limitations

The retrospective, single-center design may limit generalizability. Our reliance on routinely documented ED indicators potentially introduced inconsistency and residual confounding, in addition to the lack of comorbidity profiles and psychosocial variables. Detailed information on exposure setting, comprehensive vital signs, and metabolic state were not systematically addressed. We only assessed short-term outcomes, and the limited sample size may also affect the statistical reliability. Furthermore, we did not capture in-hospital mortality beyond the ED nor detailed causes of death. Frailty, comorbidities, potential intoxicants, specific details of exposure, and the workflow of the local emergency department/intensive care unit capacity can all change the direction of correlations. Such factors can not only impact the way triage or temperature are modulated but also influence how a model can be generalized and calibrated from one system to another. Even though these parameters were not measured, they are extremely important in both mortality and intensive care unit admission assessment. The triage categorization of patients with a tympanic temperature of less than 32°C may have introduced collinearity, regardless of our statistical effort.

The use of monthly average temperatures may have resulted in oversimplified weather patterns, potentially resulting in the absence of finer meteorological effects. Despite standardization, tympanic thermometry may underestimate true core temperature in profoundly hypothermic patients compared with central measurements, although the thermometer used fulfils current ASTM accuracy criteria. Of the 131 readings, 14 lay within 0.3 °C of the 24/28/32/35 °C boundaries, and 13 within 20–24 °C. The thermometer we used has a declared accuracy of 0.3 °C below 35 °C, these cases lie exactly within the device's tolerance range. Because of this, patients may have been classified into adjacent Swiss or WMS threshold categories purely due to measurement inaccuracy, which should be considered when interpreting analyses. While we

acknowledge the use of tympanic temperature measurement as a limitation of our study—given that it may have influenced the classification of some patients between the Swiss and WMS categories—we emphasize that our study was designed to reflect real-world triage conditions, where tympanic thermometry is the most used method in emergency departments and general clinical practice. Tympanic temperature was measured directly instead of applying the full Swiss or WMS criteria to estimate core temperature and evaluate the scales' utility. We relied solely on the temperature ranges specified by those scales, and thus we cannot draw definitive conclusions about the usefulness of core temperature estimation systems. We only analyzed the Hungarian Emergency Triage System, which may limit broader comparability. We did not assess additional ED workflow factors, such as waiting times or resource utilization. Generalizability may vary with triage protocols and ICU capacity, which influence admission thresholds and the composite outcome. External validation across health systems is required.

## Conclusion

Accidental hypothermia continues to pose a substantial clinical challenge in Hungary and other temperate regions, necessitating the implementation of specialized prehospital and in-hospital care. Although the temperature ranges of Swiss staging model and the Wilderness Medical Society guidelines can be beneficial in the assessment of severity, our results suggest that triage alone offers moderate discrimination for early critical outcomes. In terms of critical outcomes, deeper integration of admission temperature on arrival into triage decisions can serve as a simple, accurately calibrated risk indicator and prepare for early escalation decisions. However, external validation and local recalibration are required before routine use. In future studies, severity scales can be fine-tuned by adding dynamic temperature thresholds.

## Supporting information

**S1 Table. Illustration of the key parameters of the Hungarian Emergency Triage System (MSTR).** Examples of chief complaints, primary and secondary modifiers by triage category [11, 12].
(PDF)

**S2 Table. Common examples of vital signs and laboratory-based modifiers for MSTR categorization [11, 12].**
(PDF)

**S3 Table. Technical specifications and declared accuracy limits of the Braun ThermoScan PRO 6000 infrared tympanic thermometer [18].**
(PDF)

**S4 Table. Monthly mean ambient temperatures (°C) for months with at least one hypothermia admission, 2020–2024 (Budapest, Hungary) [19].**
(PDF)

**S5 Table. Logistic regression models for critical outcome (univariate and combined models).** The triage category and the tympanic temperature on admission are independently significant predictors of critical outcome. In the combined model, both retain their effect, while collinearity is not present. Model-level discrimination (AUC) improves from 0.683 to 0.740 when moving from triage to triage + temperature, and explanatory power (Nagelkerke $R^2$) also increases.
(PDF)

**S1 Fig. Calibration of the combined triage + temperature model for the composite critical outcome (ICU admission or ED mortality).** Panel A: Calibration plot across risk deciles (observed vs predicted event rates). Panel B: Reliability curve with LOWESS smoothing. The dashed line represents perfect calibration (slope = 1, intercept = 0).
(TIFF)

**S6 Table. Predictive performance of triage category, admission temperature, Swiss and WMS stages, and combined triage + temperature for composite critical outcome (ICU admission or ED mortality).** Discrimination indices (AUC, sensitivity, specificity, likelihood ratios, predictive values) are shown at the Youden-optimal threshold for each predictor. Calibration indices (Brier score, intercept, slope) refer only to the triage + temperature model, which achieved the highest AUC value among the tested predictors. Abbreviations: AUC, area under the ROC curve; LR, likelihood ratio; PPV, positive predictive value; NPV, negative predictive value.
(PDF)

**S7 Table. Collinearity diagnostics and sensitivity analyses.** There is a moderate but low-risk relationship between triage and tympanic temperature, collinearity is not confirmed.
(PDF)

## Author contributions

**Conceptualization:** Kornél Ádám, Csaba Varga.

**Data curation:** Kornél Ádám, Anna Stelkovics, Barbara Zadravecz-Heider, Barbara V. Farkas.

**Formal analysis:** Kornél Ádám.

**Investigation:** Anna Stelkovics, Barbara Zadravecz-Heider.

**Methodology:** Dóra Melicher, Csaba Varga.

**Supervision:** Dóra Melicher, Bánk G. Fenyves, Szabolcs Gaál-Marschal, Csaba Varga.

**Validation:** Bánk G. Fenyves, Csaba Varga.

**Visualization:** Kornél Ádám, Anna Stelkovics.

**Writing – original draft:** Kornél Ádám, Bánk G. Fenyves, Csaba Varga.

**Writing – review & editing:** Kornél Ádám, Dóra Melicher, Zsolt Bognár, Barbara V. Farkas, Bánk G. Fenyves, Szabolcs Gaál-Marschal, Csaba Varga.

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
