## [Decision Letter · Decision Letter 0]

5 Sep 2025

Dear Dr. Ádám,

We look forward to receiving your revised manuscript.

Kind regards,

Chiara Lazzeri

Academic Editor

PLOS ONE

Journal Requirements:

2. We note that there is identifying data in the Supporting Information file “S4_Table.pdf”. Due to the inclusion of these potentially identifying data, we have removed this file from your file inventory. Prior to sharing human research participant data, authors should consult with an ethics committee to ensure data are shared in accordance with participant consent and all applicable local laws.

-Location data

Reviewers' comments:

Reviewer's Responses to Questions

**Comments to the Author**

1. Is the manuscript technically sound, and do the data support the conclusions?

Reviewer #1: Yes

Reviewer #2: Partly

2. Has the statistical analysis been performed appropriately and rigorously?

Reviewer #1: Yes

Reviewer #2: I Don't Know

3. Have the authors made all data underlying the findings in their manuscript fully available?

Reviewer #1: Yes

Reviewer #2: Yes

4. Is the manuscript presented in an intelligible fashion and written in standard English?

Reviewer #1: Yes

Reviewer #2: Yes

Reviewer #1: In this reviewer's opinion, the manuscript addresses an important and understudied area by exploring seasonal patterns and predictors of critical outcomes in patients with accidental hypothermia. The research is well-structured, and the analyses are thorough. However, there are several areas that require clarification and further detail to strengthen the manuscript before publication. Here are my major comments-

1. Potential collinearity between predictors

The Hungarian Emergency Triage System (MSTR) classification and admission core temperature were both analyzed as predictors of critical outcomes (ICU admission or ED mortality). However, MSTR triage categories already incorporate core temperature thresholds (e.g., core temperature <32 °C is a modifier for a higher urgency category, lines 59–61). This introduces the possibility of collinearity, as the predictors may be highly correlated. Please clarify whether collinearity diagnostics beyond variance inflation factor (VIF) were performed (lines 125–128 mention VIF briefly). If not, additional analyses or sensitivity checks should be considered.

2. Seasonal variation versus ambient temperature association

The study observed a clear seasonal variation in hypothermia cases, with higher incidence during colder months (lines 162–172 and Figure 2). However, environmental (ambient) temperature was not independently associated with critical outcomes in univariate or multivariate analysis (lines 186–188, 260–262). This raises an important interpretive question: why do hypothermia cases cluster seasonally, yet ambient temperature does not predict severity or mortality? Please expand the discussion to provide context and potential explanations for this divergence. This would also underscore that preventive strategies should target vulnerable populations year-round, not only during cold spells.

3. Figure legends require more detail for standalone interpretation

Figures 2 and 3 (lines 167–172 and 218–221) are central to the manuscript’s findings, but the legends are too brief and do not allow readers to interpret the figures independently. Suggested improvements:

* Figure 2: Please specify the sample size (n=131), define each Swiss staging category (I–IV), clarify that ambient temperatures are monthly means, and indicate the source of meteorological data (reference 19).

* Figure 3: The legend should define all abbreviations (e.g., “triage category,” “Swiss staging model,” “WMS classification”), clarify the outcome being predicted (“critical outcome” = ICU admission or ED mortality), and explain the meaning of AUC values shown. Please also ensure consistent terminology throughout (“critical outcome” vs. “critical resolution”).

4. Expand on unmeasured variables in the limitations section

The limitations section mentions the absence of comorbidity, frailty, and psychosocial data. I recommend expanding this discussion to explain how these unmeasured variables might bias the observed associations, particularly regarding mortality and ICU admission. Acknowledging their potential confounding effects will improve the transparency and interpretability of the study’s predictive findings.

5. Tympanic thermometry and risk of misclassification

The discussion correctly notes that tympanic thermometry may underestimate core temperature. Please specify how many patients had readings near the lower detection limit and whether this may have affected classification into the Swiss or WMS severity categories (lines 280–286, 150–153, 105–108). Clarifying this will help readers understand the potential for misclassification bias.

6. Generalizability to other settings

As this is a single-center study, please add a brief statement on how differences in triage protocols or ICU capacity in other healthcare systems might affect the applicability of these findings (lines 269–275, 290–292).

Addressing these points will significantly improve the clarity, robustness, and generalizability of the manuscript. I appreciate the authors’ efforts in conducting this important study and look forward to seeing the revised version.

Reviewer #2: Thanks for this submission and the opportunity.

I feel there are too many directions to this piece.

The title makes it seem that tracking seasonal patterns is the main subject but that is in fact entirely covered by a single line in the results and then an un-discussed Fig 2 which to me does show some differences in years.

Then there is a lot of stats - too much and thinly described in my opinion - and a nebulous outcome on whether triage is picking up hypothermia adequately or not.

I'd be interested in the presentation, management and complications in the 130 cases you describe - but the only information you provide is the length of stay in the ED which leaves me questioning.

The conclusion to me says further research and not much more.

**Do you want your identity to be public for this peer review?** For information about this choice, including consent withdrawal, please see our Privacy Policy

Reviewer #1: **Yes: ** Awsaf Karim

Reviewer #2: No

---

## [Author Response · Author response to Decision Letter 1]

18 Sep 2025

Point-by-Point Response to Reviewers

We thank the Academic Editor and both reviewers for their thorough evaluation of our manuscript. We have addressed each comment in detail. Reviewer comments are reproduced in quotes, followed by our responses. Line number references correspond to the revised manuscript.

Reviewer #1 (relevant parts in the manuscript are highlighted in pink)

Comment 1

“1. Potential collinearity between predictors.

The Hungarian Emergency Triage System (MSTR) classification and admission core temperature were both analyzed as predictors of critical outcomes (ICU admission or ED mortality). However, MSTR triage categories already incorporate core temperature thresholds (e.g., core temperature <32 °C is a modifier for a higher urgency category, lines 59–61). This introduces the possibility of collinearity, as the predictors may be highly correlated. Please clarify whether collinearity diagnostics beyond variance inflation factor (VIF) were performed (lines 125–128 mention VIF briefly). If not, additional analyses or sensitivity checks should be considered.”

Response: We appreciate your insightful comment. In the updated paper, we performed additional diagnostics to examine this possible overlap and clarified the results. We calculated both Pearson's correlation (r) and Spearman's rank correlation (p) between the MSTR triage category and the admission tympanic temperature. The results showed that the association between the two parameters is only modest. We also looked at the condition index of the model, which, in this case, is indicative of low multicollinearity. The outcomes mentioned above have been presented in S7 Table. In the Methods section, we have changed the text accordingly. We have noted in the Discussion that these diagnostics were not showing any major collinearity. Moreover, there is now a sentence in the Discussion stating that in our cohort the admission temperature and triage category are only modestly correlated and thus the combination of these two factors allows for better prediction (as shown by the improved model performance). In addition to VIFs, we computed the condition index, which was 1.31 in the combined model, further indicating low multicollinearity.

Comment 2:

“2. Seasonal variation versus ambient temperature association.

The study observed a clear seasonal variation in hypothermia cases, with higher incidence during colder months (lines 162–172 and Figure 2). However, environmental (ambient) temperature was not independently associated with critical outcomes in univariate or multivariate analysis (lines 186–188, 260–262). This raises an important interpretive question: why do hypothermia cases cluster seasonally, yet ambient temperature does not predict severity or mortality? Please expand the discussion to provide context and potential explanations for this divergence. This would also underscore that preventive strategies should target vulnerable populations year-round, not only during cold spells.”

Response: We thank the reviewer for highlighting this point. The paradox was recognized in our original manuscript, but we did not really go into sufficient detail to explain it. Therefore, we have now widened the Results and Discussion sections to deal with both the seasonal incidence of the disease and the apparent lack of a direct impact of ambient temperature on outcomes. We propose that patient-level and situational factors, such as exposure duration, frailty, comorbidities, intoxication, indoor vs outdoor setting, and prehospital circumstances modulate the relationship between weather and outcome severity. We also added a note on prevention, in line with the reviewer’s suggestion to emphasize that although winter has more cases, hypothermia can also occur in milder conditions, and vulnerable groups require attention throughout the year. We hope these additions clarify the issue and strengthen the paper.

Comment 3:

“3. Figure legends require more detail for standalone interpretation.

Figures 2 and 3 (lines 167–172 and 218–221) are central to the manuscript’s findings, but the legends are too brief and do not allow readers to interpret the figures independently. Suggested improvements:*

• Figure 2: Please specify the sample size (n=131), define each Swiss staging category (I–IV), clarify that ambient temperatures are monthly means, and indicate the source of meteorological data (reference 19).

• Figure 3: The legend should define all abbreviations (e.g., “triage category,” “Swiss staging model,” “WMS classification”), clarify the outcome being predicted (“critical outcome” = ICU admission or ED mortality), and explain the meaning of AUC values shown. Please also ensure consistent terminology throughout (“critical outcome” vs. “critical resolution”).”*

Response: We appreciate these detailed suggestions. We have revised both figure legends extensively to ensure they are self-explanatory and incorporate all the recommended details. Here are the specific changes made:

• Figure 2 Legend (Seasonal distribution of hypothermia cases): We have expanded this legend to a full paragraph that now reads:

“Fig 2. Seasonal distribution of hypothermia cases and ambient temperatures (2020–2024). Bars indicate monthly case counts stratified by Swiss staging (I: 35–32 °C; II: 32–28 °C; III: 28–24 °C; IV: <24 °C); the line shows the monthly mean outdoor temperature for patients’ registered area. Sample size: n = 131. Meteorological source: monthly mean temperatures from the local official weather station (see S4 Table).”

These additions address all points, we explicitly state “Sample size: n = 131” so readers know how many cases the figure encompasses. We define each Swiss staging category (I–IV) with their corresponding core temperature ranges in °C. This way, even without referring back to Table 1 or the text, a reader can understand what I, II, III, IV mean on the bar colors. We clarify that the plotted temperatures are monthly mean outdoor temperatures for the region (and not individual patient temperatures or daily highs, etc.). Using the phrase “monthly mean outdoor temperature” makes this clear. We cite the source of the meteorological data, which is our reference [19] (a Hungarian meteorological database). Instead of giving the full reference in the legend, we noted in brief that the data come from the local station and pointed to S4 Table where we list those monthly values. In S4 Table (Supporting Information), we have the month-by-month mean temperatures listed, and reference [19] in the main reference list is the source. This should satisfy the need for attribution of data source.

• Figure 3 Legend (ROC curves for predictors of critical outcome): We have also rewritten this legend to incorporate all requested clarifications. The updated legend now reads:

“Fig 3. ROC curves for predictors of the composite critical outcome (ICU admission or ED mortality). Curves are shown for MSTR triage category, admission tympanic (core‑proxy) temperature (°C), Swiss staging, Wilderness Medical Society (WMS) classification, and the combined triage + temperature model. AUC values summarize overall discrimination (higher is better). Abbreviations: AUC, area under the ROC curve; MSTR, Hungarian Emergency Triage System; WMS, Wilderness Medical Society.”*

We clearly state what outcome is being predicted by these ROC curves: “the composite critical outcome (ICU admission or ED mortality).” This was crucial, as a reader should know that the ROC/AUC pertains to predicting the composite outcome defined in our study. We use the term “critical outcome” here and explicitly define it in parentheses so there is no ambiguity. We define all abbreviations in the legend: MSTR is spelled out as the Hungarian Emergency Triage System (our triage scheme). WMS is spelled out as Wilderness Medical Society classification. The AUC is defined as area under the ROC curve. We also included “(core-proxy)” next to “tympanic temperature” to quickly indicate that this is a core temperature proxy measure. We explain the meaning of the AUC values in general terms: “AUC values summarize overall discrimination (higher is better).” This sentence ensures that even a reader not deeply familiar with ROC analysis will understand that a higher AUC means a more accurate predictor. We have removed the term “critical resolution” entirely from the manuscript and replaced it with “critical outcome” wherever applicable. We corrected this in the revised manuscript, and this should eliminate any confusion. Thank you for pointing this out.

Comment 4:

“4. Expand on unmeasured variables in the limitations section.

The limitations section mentions the absence of comorbidity, frailty, and psychosocial data. I recommend expanding this discussion to explain how these unmeasured variables might bias the observed associations, particularly regarding mortality and ICU admission. Acknowledging their potential confounding effects will improve the transparency and interpretability of the study’s predictive findings.”

Response: We agree that potential impact of the undiscussed confounders is important as it may affect the balance of the interpretation of our results. Initially, we only had a brief mention of the missing data on comorbidities and social factors. We added text to the ‘Limitation’ section, explaining how the lack of these data might bias our results. We note that leaving out these factors could introduce confounding and tends to bias associations. We also added that these factors influence generalizability and calibration of any predictive model, also making it clear to the reader that we recognize the predictors in our study do not operate in a vacuum. Thank you for this recommendation. We believe this change significantly improves the manuscript’s balance and honesty about its limitations.

Comment 5:

“5. Tympanic thermometry and risk of misclassification.

The discussion correctly notes that tympanic thermometry may underestimate core temperature. Please specify how many patients had readings near the lower detection limit and whether this may have affected classification into the Swiss or WMS severity categories (lines 280–286, 150–153, 105–108). Clarifying this will help readers understand the potential for misclassification bias.”

Response: Misclassification due to the limitations of tympanic temperature measurement is an important issue to quantify. We have added specific details to the ‘Limitations’ to directly address this point. The device we used has a measurement range of 20.0 °C to 42.2 °C. We did not have any readings below 20 °C (the absolute lower detection limit of the device). On the other hand, we identified 13 patients with readings in the 20–24 °C range, close to the device’s technical lower limit. Fourteen patients’ temperatures fell within ±0.3 °C of key Swiss/WMS thresholds. Given the thermometer’s stated accuracy, some of these patients may have been classified into adjacent severity stages purely due to measurement error. We acknowledge this as a methodological limitation, but emphasize that tympanic thermometry reflects real-world ED practice, and we quantified the extent of possible misclassification to aid interpretation.

Comment 6:

“6. Generalizability to other settings

As this is a single-center study, please add a brief statement on how differences in triage protocols or ICU capacity in other healthcare systems might affect the applicability of these findings (lines 269–275, 290–292).”

To address the reviewer's comment on generalizability, we clarified the need for external validation, acknowledging the effect of different distributions of ICU resources and triage algorithms that may lead to different levels of transferability to another hospital or region.

We hope the above responses satisfactorily address all of Reviewer #1’s comments. We have implemented all suggested changes in the manuscript and believe the manuscript is much improved as a result. Thank you for your thorough review and positive feedback on the importance and structure of our study.

Reviewer #2 (relevant parts in the manuscript are highlighted in blue)

Comment 1:

“Thanks for this submission and the opportunity. I feel there are too many directions to this piece. The title makes it seem that tracking seasonal patterns is the main subject but that is in fact entirely covered by a single line in the results and then an un-discussed Fig 2 which to me does show some differences in years.”

Response: Thank you for your comment. In the revised manuscript, we modified the title, removing the reference to “seasonal patterns.” This makes it more accurately reflect the focus of the manuscript and eliminates the contradiction that, despite the title referring to seasonal trends, this only plays a minor role in the content. The text has also been expanded: in the Results section, the description of Figure 2 has been explained in more detail, and in the Discussion section, the significance of seasonal variation has been fine tuned. With these changes, we believe that the title and content are now consistent.

Comment 2:

“Then there is a lot of stats - too much and thinly described in my opinion - and a nebulous outcome on whether triage is picking up hypothermia adequately or not.”

Response: The statistical analyses have been restructured and condensed to make them more readable for the reader. In the Results section and Discussion, now it is more clearly highlighted which predictors performed well. In particular, the revised version more clearly describes that triage category alone is a better predictor of critical outcome than hypothermia severity scales, and that the inclusion of admission tympanic temperature further improves prediction. These results have been summarized in a user-friendly manner, for example, describing the sensitivity, specificity, and calibration of the combined model at the optimal cutoff. Overall, the statistical part has become more transparent and better supports the main message. Thank you for pointing this out, it prompted us to sharpen the take-home messages from our analysis.

Comment 3:

“I'd be interested in the presentation, management and complications in the 130 cases you describe - but the only information you provide is the length of stay in the ED which leaves me questioning.”

Response: We agree with the reviewer that it would be interesting and important to present the clinical background of hypothermia cases in more detail and we acknowledge that this kind of information is still scarce. This is important additional information, but actual clinical characteristics (e.g. condition on arrival, resuscitation/rewarming interventions used, complications) cannot be explained in detail. We now clearly state this in ‘Limitations’ that detailed data on comorbidities, exposure conditions, vital parameters and social factors were not available.

Comment 4:

“The conclusion to me says further research and not much more.”

Response: We apologize that our original conclusion was not as substantive as it should have been. In the revised manuscript, we have rephrased the Conclusion section to ensure it provides clear, concrete take-home messages rather than a vague call for further research, while still avoiding overstatements. We recognize that accidental hypothermia is an important clinical problem in Hungary and other regions with moderate climate, and this is exactly the reason our study has such an important role. The thresholds of hypothermia stages according to Swiss and WMS guidelines are still considered as a valuable reference, our results show that general triage alone has moderate discrimination for early critical outcomes. Most importantly, the use of admission temperature as part of triage offers a straightforward, well-calibrated risk indication that can be very helpful in the decision of early escalation. This provides a simple and practical guide for early clinical decision-making. We also recognize the necessity of external validation and some local recalibration, as the future work suggested would be to explore the change of temperature thresholds as a method of refining severity scales. We are thankful to Reviewer #2 for the suggestion to sharpen our focus and conclusions. The remarks made it possible for us to i

---

## [Editor Report · Decision Letter 1]

26 Sep 2025

Accidental hypothermia in emergency care: multifactorial triage-based prediction of early critical outcomes in a temperate-climate cohort

PONE-D-25-33648R1

Dear Dr. Ádám,

We’re pleased to inform you that your manuscript has been judged scientifically suitable for publication and will be formally accepted for publication once it meets all outstanding technical requirements.

Kind regards,

Chiara Lazzeri

Academic Editor

PLOS ONE
---

## [Editor Report · Acceptance letter]

PONE-D-25-33648R1

PLOS ONE

Dear Dr. Ádám,

I'm pleased to inform you that your manuscript has been deemed suitable for publication in PLOS ONE. Congratulations! Your manuscript is now being handed over to our production team.

Kind regards,

on behalf of

Dr. Chiara Lazzeri

Academic Editor

PLOS ONE